# Clinical impact of angiographically insignificant suboptimal poststent findings detected by optical coherence tomography after drug-eluting stent implantation

Jae Young Cho[ID][1‡], Hyungdon Kook[ID][2‡], Cheol Woong Yu[ID][2]*

1 Division of Cardiology, Department of Internal Medicine, Wonkwang University Hospital, Iksan, Korea,
2 Division of Cardiology, Department of Internal Medicine, Korea University Anam Hospital, Seoul, Korea

‡ These authors share first authorship on this work.
* ycw717@naver.com

**Data Availability Statement:** All relevant data are within the manuscript and its Supporting Information file.

## Abstract

### Background

Although optical coherence tomography (OCT)-detected suboptimal findings (SF-OCT) such as malapposition, edge dissection, tissue protrusion, thrombus and small minimal stent area (MSA) are frequently observed after the implantation of drug-eluting stents (DES), their clinical implications are controversial.

### Hypothesis

Clinical outcomes may differ between patients with SF-OCTs and without SF-OCTs after DES implantation.

### Methods

A total of 576 patients undergoing OCT analysis after DES implantation were divided into SF-OCT group (n = 379, 379 lesions) and No SF-OCT group (n = 197, 197 lesions). The study population had no significant abnormal finding in final angiography. Quantification was performed for each SF-OCT. The incidences of major adverse cardiovascular events (MACE: all-cause death, non-fatal myocardial infarction, target vessel revascularization, and stent thrombosis) were compared between the two groups. A median follow-up duration was 21.5 months.

### Results

Among 379 patients with SF-OCT, 32.4% had multiple SF-OCTs. Malapposition (32.1%, IQR of maximal depth 315–580 μm) was the most frequent, followed by small MSA (31.6%), edge dissection (12.5%, IQR of maximal flap of opening 0.27–0.52 mm), thrombus (7.6%, IQR of diameter 1.31–1.97mm) and tissue protrusion (6.8%, IQR of diameter 1.05–1.67 mm). The SF-OCT group showed smaller stent diameter and longer stent length, and lower

**Funding:** Jae Young Cho received funding from Wonkwang University [2018-091474]. The funder had no role in study design, data collection and analysis, decision to publish, or preparation of the manuscript.

**Competing interests:** The authors have declared that no competing interests exist.

in-stent lumen expansion rate. The incidence of MACE did not differ between the two groups (3.0% for No SF-OCT vs. 5.0% for SF-OCT; HR 1.601; 95% CI 0.639 to 4.011; P = 0.310).

## Conclusions

The presence of angiographically insignificant SF-OCTs were not associated with clinical outcomes in this study.

## Introduction

Recent reports of optical coherence tomography (OCT) analysis after drug-eluting stent (DES) implantation revealed that suboptimal findings detected by OCT (SF-OCT) had a high prevalence, while no significant abnormal findings were noted on conventional coronary angiography. The SF-OCTs have been reported to be malapposition, tissue protrusion (TP), edge dissection (ED), thrombus and small minimal stent area (MSA) in previous studies [1–6]. However, their clinical implications are controversial. Clarifying the relationships between SF-OCTs and adverse clinical outcomes could help improve clinical outcomes and avoid unnecessary additional procedures after percutaneous coronary intervention (PCI).

Recent studies of SF-OCT focused predominantly on the natural course of these findings, demonstrating that most of them except for malapposition improved or resolved within one year [1–5]. However, data regarding its impact on relevant clinical outcomes is controversial. In addition, there has not been comparative studies with patients without SF-OCT. Therefore, we compared clinical outcomes after DES implantation between patients with or without suboptimal findings differentiated by OCT.

## Methods

### Study patients

The Optical Coherence Tomography Registry of Korea University Anam Hospital is a single-center registry of patients undergoing OCT imaging of coronary arteries (ClinicalTrials.gov Trial Number: NCT02966262; URL: https://clinicaltrials.gov/ct2/show/NCT02966262?term=NCT02966262&draw=2&rank=1). A total of 576 patients in the OCT registry was retrospectively reviewed for the study at Cardiovascular Center, Korea University Anam Hospital from January 2011 to May 2013. Patients were included into the study based on the following criteria: 1) Patients who had only single lesion to intervene; 2) Patients who underwent sequential OCT immediately after DES implantation or after adjuvant procedures; 3) No significant abnormal finding in final coronary angiography. Patients were allocated to either SF-OCT group or No SF-OCT group, based on whether they exhibited at least one SF-OCT after OCT analysis. After stent implantation, all patients received dual antiplatelet therapy unless contraindicated. The therapy was maintained for at least 12 months.

Demographic data, including sex, age, body mass index, comorbidities, prescribed drugs, laboratory data, and clinical presentation, were collected and compared between the two groups. Left ventricular dysfunction was defined as a left ventricular ejection fraction <45%. The primary endpoint was major adverse cardiovascular events (MACE), defined as a composite of cardiac death, non-fatal myocardial infarction (MI), the need for repeated target vessel revascularization (TVR), and stent thrombosis. TVR was ischemia-driven. The secondary endpoint was each component event of MACE. Information of clinical outcomes was collected by the retrospective review of the chart. Collecting data of clinical information and outcomes was

performed blindly to angiographic data and OCT findings. This study was approved by the Korea University Hospital Institute Review Board (IRB No. 2016AN0095), and the informed consent was waived due to retrospective study design. This study also complied with the Declaration of Helsinki.

## Angiographic analysis

Coronary angiograms were analyzed using a computer-based Telecardiology system, version 2.02 (Medcon Inc., Tel Aviv, Israel) by three radiologic technicians who were blinded to the study purpose. The reference diameter, minimal luminal diameter (MLD), percentage of stenosis, and lesion length were evaluated from diastolic frames using guided catheter magnification calibration in a single, matched view with a computerized quantitative analyzer using a caliper. The average diameter of normal segments proximal and distal to the treated lesion was used as the reference diameter.

## OCT acquisition

OCT examination and analysis were performed immediately after stent implantation (Light-Lab Imaging Inc., Ilumien Offline review workstation, Ver E.4.1, MA, USA). Using a 0.014" guide wire, an OCT imaging catheter (C7 Dragonfly$^{TM}$, LightLab Imaging Inc., MA, USA) was advanced into the distal end of the DES implantation site. The entire length of the stent was imaged with an automatic pullback device moving at 15 mm/s. The whole stent was clearly visualized on each OCT image; in-segment cross-sectional views were also obtained.

## OCT analysis

All baseline OCT images were reviewed by an independent observer who was blinded to the clinical presentation, lesion, and procedural characteristics. The analysis encompassed the intra-stent segment, defined by the first and the last cross-sections with a visible strut, and the adjacent vessel segments 5 mm proximal and distal to the stent, defined as edge segments. Quantification was performed for each finding.

TP was defined as tissue protruding between adjacent stent struts toward the lumen, with or without disruption of luminal vessel surface continuity (Fig 1A) [7]. TP was distinguished from thrombus by visualizing behind the plaque. TP length was defined as the longest diameter of protruded tissue. The area of TP was also measured (tissue protrusion area).

ED was defined as disruption of the luminal vessel surface in the edge segments (within 5 mm proximal and distal to the stent, with no visible struts) without flow limiting (Fig 1B) [7]. Maximum flap length (from its tip to the joint point with the vessel wall), maximum flap opening (distance from the flap tip to the lumen contour along a line projected through the gravitational center of the lumen), arc of circumferential extension, and longitudinal flap length were measured in the cross-sectional images [8].

Stent malapposition was defined as a clear separation between at least one stent strut and the strut reflection, in addition to a vessel wall +20 μm greater than the actual stent thickness on OCT images (Fig 1C) [7, 9]. The maximum distances from the endoluminal surface of the strut to the vessel wall (maximum depth), the malapposition area (difference between lumen area and stent area), and the longitudinal distance of malapposition were measured.

Intracoronary thrombus was defined as a mass protruding beyond the stent strut into the lumen with significant attenuation behind the mass [10, 11]. The presence of a thrombus was assessed quantitatively, and the diameter of the visible thrombus was recorded (Fig 1D).

We defined a small MSA as an in-stent minimum area <4.5 mm$^2$ and underexpansion as MSA <80% of the average reference lumen area [6, 12].

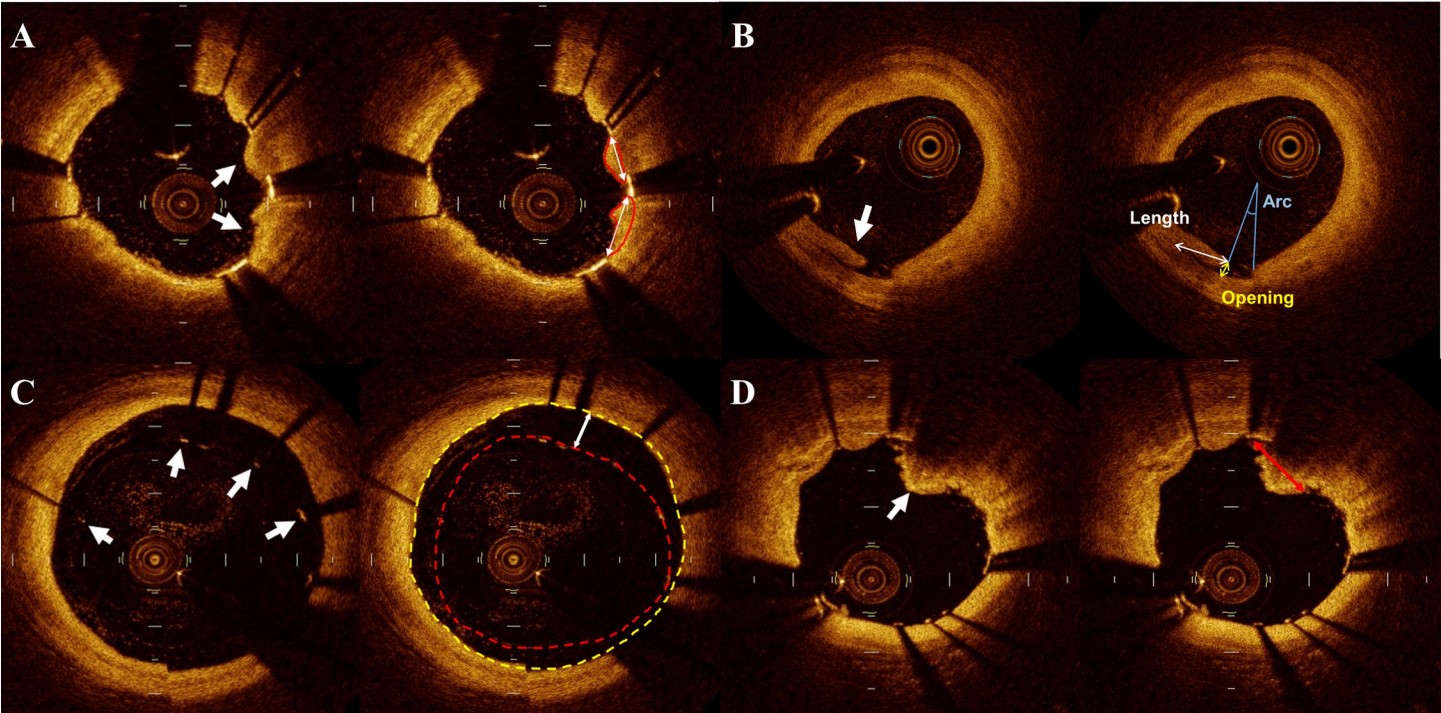

**Fig 1. Representative images of suboptimal findings detected on optical coherence tomography.** (A) Tissue protrusion (white arrows). Maximal length (white two-way arrows) and area (red line) of tissue protrusion. (B) Edge dissection (white arrow). Maximal flap opening (yellow two-way arrow), maximal length (white two-way arrow) and arc (blue line) of edge dissection. (C) Malapposition (white arrows). Lumen area (yellow broken line), stent area (red broken line) and maximal depth (white two-way arrow) of malapposition. Malapposition area was calculated by subtracting stent area from lumen area. (D) Thrombus (white arrow). Diameter (red two-way arrow) of thrombus.

For clinical impact, only previously noted significant findings were used for analysis and the following factors were considered significant SF-OCTs [3, 6]:

1. TP: diameter ≥0.5mm and protrusion area/stent area at site of tissue protrusion ≥10%

2. ED: maximum flap opening ≥0.2mm

3. Malapposition: maximum depth ≥200μm

4. Thrombus: diameter ≥0.5mm and thrombus area/stent area at site of thrombus ≥10%

5. Small MSA: in-stent minimum area <4.5 mm$^2$

If adjuvant procedures, such as balloon dilation or stent implantation, were performed based on OCT by physician's preference, SF-OCTs were analyzed after the adjuvant procedure. OCT-based quantitative measurements were assessed according to well-standardized methods.

## Statistical analysis

Data are expressed as mean± standard deviation for continuous variables, whereas data for categorical variables are expressed as number and percentage of patients. The chi-square test was used to compare categorical variables. Event rates were estimated using Kaplan-Meier survival analysis at 5 years, and hazard ratios (HR) were generated using Cox regression analysis. Because patients may have experienced more than 1 MACE, each patient was assessed until

the occurrence of his or her first event and only once during analysis. To determine the association between clinical characteristics and outcomes, univariate and multivariate Cox regression analyses were performed for the entire population. The Cox regression model included the following variables, which were considered to be related with clinical outcomes: age, body mass index, hypertension, diabetes mellitus, clinical diagnosis, smoking history, lipid profile, peak CK-MB, use of renin-angiotensin system blocker and beta blocker, stent diameter, stent length, MSA, proximal and distal reference area, underexpansion, SF-OCT, components and quantitative measurements of SF-OCT. Additionally, selected variables were tested for logistic univariate regression associated with SF-OCTs; if P-value <0.05, they were simultaneously entered into a logistic multivariate regression model to identify independent predictors of SF-OCTs and to calculate their adjusted odds ratios (OR) with associated 95% confidence intervals (CI). The logistic regression model included the following variables, which were considered to be related with SF-OCTs: age, male sex, hypertension, diabetes mellitus, clinical diagnosis, lipid profile, peak CK-MB, stent diameter, stent length, MSA, proximal reference area, distal reference area, underexpansion and adjuvant procedure. SPSS version 20.0 (IBM SPSS Statistics, IBM Corporation, Armonk, New York) was used for all analyses. A P-value <0.05 was considered statistically significant.

## Results

The study protocol is diagrammed in Fig 2. Overall, a total of 576 patients with 576 lesions who underwent OCT immediately after DES implantation was analyzed. Adjuvant procedures after OCT examination were performed in 132 patients (132 lesions). Among overall study population, 379 patients (379 lesions) showed one or more SF-OCT (SF-OCT group) and 197 patients (197 lesions) showed no SF-OCT in the final OCT examination (No SF-OCT group). The baseline characteristics were not different between the No SF-OCT group versus the SF-OCT group (Table 1). Table 2 shows the quantitative coronary angiography results. Smaller stent diameter (2.98±0.42 vs. 3.27±0.43 mm, P <0.001) and longer stent length was used (23.24±7.56 vs. 20.53±7.20 mm, P <0.001) in the SF-OCT group. MSA (5.27 ± 2.02 vs. 6.85 ± 2.08 mm$^2$, P<0.001) and percentage of in-stent lumen expansion (72.53 ± 11.06 vs. 78.95 ± 9.24%, P<0.001) were also smaller in the SF-OCT group. The incidence of underexpansion was more frequent in the SF-OCT group (77.3% vs. 51.3%, P<0.001).

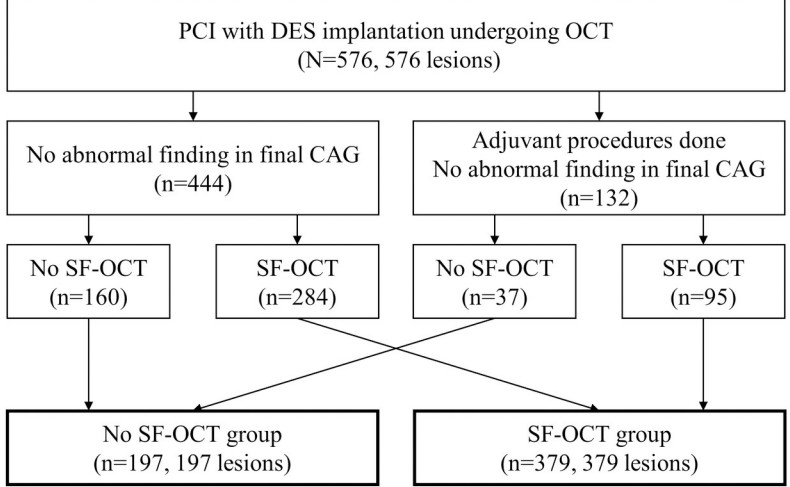

**Fig 2. Study flow chart.** PCI: percutaneous coronary intervention DES: drug-eluting stent; OCT: optical coherence tomography; CAG: coronary angiography; SF-OCT: optical coherence tomography detected suboptimal findings.

**Table 1. Baseline patient characteristics.**

| Variable | No SF-OCT | SF-OCT | *P*-value |
|---|---|---|---|
| | (197 patients) | (379 patients) | |
| Age (year) | 61.2 ± 11.8 | 62.0 ± 10.9 | 0.415 |
| Male sex (%) | 142 (72.1) | 289 (76.3) | 0.312 |
| Body mass index (kg/m$^2$) | 24.7 ± 3.1 | 24.9 ± 3.1 | 0.775 |
| Smoking, n (%) | | | |
| Previous | 22 (11.2) | 66 (17.4) | 0.057 |
| Current | 57 (28.9) | 121 (31.9) | |
| Comorbidity | | | |
| Hypertension, n (%) | 109 (55.3) | 240 (63.3) | 0.072 |
| Diabetes mellitus, n (%) | 59 (29.9) | 145 (38.3) | 0.054 |
| CVA history, n (%) | 6 (3.0) | 15 (4.0) | 0.647 |
| Left ventricular dysfunction, n (%) | 12 (6.1) | 26 (6.9) | 0.860 |
| Previous PCI history, n (%) | 23 (11.7) | 44 (11.6) | 1.000 |
| Laboratory Data | | | |
| White blood cell count (x10$^3$/uL) | 8.2 ± 3.5 | 7.8 ± 3.3 | 0.274 |
| Creatinine (mg/dL) | 1.12 ± 1.31 | 1.21 ± 1.48 | 0.466 |
| Total cholesterol (mg/dL) | 168.6 ± 66 | 163.7 ± 67.4 | 0.405 |
| Triglyceride (mg/dL) | 140.6 ± 130.1 | 138.3 ± 103.8 | 0.816 |
| HDL-cholesterol (mg/dL) | 41.1 ± 13.9 | 39.0 ± 15.6 | 0.117 |
| LDL-cholesterol(mg/dL) | 110.8 ± 48.5 | 102.4 ± 53.4 | 0.066 |
| Peak CK-MB (ng/mL) | 47.7 ± 98.5 | 42.9 ± 92.7 | 0.580 |
| hs-CRP (mg/dL) | 3.4 ± 13.8 | 4.8 ± 18.8 | 0.363 |
| ESR (mm/hr) | 7.3 ± 8.5 | 8.6 ± 13.7 | 0.174 |
| Drug | | | |
| Aspirin, n (%) | 195 (99.0) | 377 (99.5) | 0.609 |
| Clopidogrel, n (%) | 195 (99.0) | 378 (99.7) | 0.271 |
| Statin, n (%) | 188 (95.4) | 366 (96.6) | 0.499 |
| RAS blocker[‡], n (%) | 105 (53.3) | 205 (54.1) | 0.387 |
| Beta blocker, n (%) | 93 (47.2) | 194 (51.2) | 0.381 |
| Calcium channel blocker, n (%) | 56 (28.4) | 112 (29.6) | 0.847 |
| Clinical presentation | | | |
| Stable angina, n (%) | 56 (28.4) | 92 (24.3) | 0.600 |
| Unstable angina, n (%) | 83 (42.1) | 187 (49.3) | |
| NSTEMI, n (%) | 26 (13.2) | 45 (11.9) | |
| STEMI, n (%) | 32 (16.2) | 55 (14.5) | |

SF-OCT: optical coherence tomography detected suboptimal findings; CVA: cerebrovascular accident; PCI: percutaneous coronary intervention; RAS: renin–angiotensin system; NSTEMI: non–ST-segment elevation myocardial infarction; STEMI: ST-segment elevation myocardial infarction

## SF-OCTs after DES implantation

The prevalence of SF-OCTs was 65.8% (379/576). The incidences of the individual SF-OCT were as follows: 185 malapposition (32.1%), 182 small MSA (31.6%), 72 ED (12.5%), 44 thrombi (7.6%), and 39 TP (6.8%) (Fig 3A). Of 379 lesions with SF-OCTs, 256 (67.5%) had one SF-OCT, 100 (26.4%) had two, 23 (6.1%) had three and none had four or five (Fig 3B). Adjuvant procedures were performed after index PCI in 132 cases (22.9%) by physician's preference, mainly by visualized significant underexpansion or malapposition in angiography or OCT. After the adjuvant procedures, 37 patients were assigned to the No SF-OCT group but

**Table 2.  Angiographic and procedural data.**

| Variable | No SF-OCT (197 lesions) | SF-OCT (379 lesions) | *P*-value |
|---|---|---|---|
| Vessel, n (%) | | | |
| LAD | 119 (60.4) | 251 (66.2) | 0.098 |
| LCX | 16 (8.1) | 39 (10.3) | |
| RCA | 56 (28.4) | 86 (22.7) | |
| Left main | 5 (2.5) | 3 (0.8) | |
| Stent type, n (%) | | | |
| Biolimus A9-eluting stent | 93 (47.2) | 179 (47.2) | 0.404 |
| Everolimus-eluting stent | 98 (49.7) | 195 (51.5) | |
| Sirolimus-eluting stent | 1 (0.5) | 2 (0.5) | |
| Bare-metal stent | 5 (2.5) | 3 (0.8) | |
| Stent diameter (mm) | 3.27 ± 0.43 | 2.98 ± 0.42 | <0.001 |
| Stent length (mm) | 20.53 ± 7.20 | 23.24 ± 7.56 | <0.001 |
| Adjuvant procedure | | | |
| Adjuvant dilatation, n (%) | 37 (18.8) | 95 (25.1) | 0.095 |
| Adjuvant balloon diameter (mm) | 3.55 ± 0.49 | 3.39 ± 0.61 | 0.137 |
| Adjuvant balloon length (mm) | 11.14 ± 4.39 | 12.03 ± 3.99 | 0.262 |
| Quantitative Coronary Analysis | | | |
| Baseline | | | |
| RD (mm) | 3.43 ± 0.43 | 3.00 ± 0.61 | 0.013 |
| MLD (mm) | 0.78 ± 0.45 | 0.64 ± 0.38 | 0.254 |
| Diameter stenosis (%) | 67.99 ± 27.78 | 64.84 ± 31.86 | 0.718 |
| Lesion length (mm) | 19.5 ± 8.5 | 22.2 ± 10.4 | 0.287 |
| Post-procedure | | | |
| RD (mm) | 3.63 ± 0.46 | 3.34 ± 0.58 | 0.092 |
| MLD (mm) | 3.35 ± 0.44 | 3.00 ± 0.46 | 0.015 |
| Diameter stenosis (%) | 5.45 ± 4.01 | 6.65 ± 5.35 | 0.402 |
| OCT Quantitative Coronary Analysis | | | |
| MSA (mm$^2$) | 6.85 ± 2.08 | 5.27 ± 2.02 | <0.001 |
| Proximal RA (mm$^2$) | 9.38 ± 2.91 | 8.06 ± 3.23 | <0.001 |
| Distal RA (mm$^2$) | 8.11 ± 2.53 | 6.61 ± 2.65 | <0.001 |
| Expansion rate (%) | 78.95 ± 9.24 | 72.53 ± 11.06 | <0.001 |
| Underexpansion, n (%) | 101 (51.3) | 293 (77.3) | <0.001 |

SF-OCT: optical coherence tomography detected suboptimal findings; LAD: left anterior descending artery; LCX: left circumflex artery; RCA: right coronary artery; RD: reference diameter; MLD: minimal lumen diameter; OCT: optical coherence tomography; RA: reference area; MSA: minimal stent area

95 patients still showed SF-OCT despite of the adjuvant procedure, thereby assigned to the SF-OCT group (Fig 2). Table 3 shows the quantitative assessment of each individual SF-OCT. Average diameter of thrombus was 1.62 mm and most of the cases showed 1 thrombus in OCT finding. In malapposition, average maximal depth was 469 μm with average malapposition area of 2.07 mm$^2$ and length of 2.43 mm. In cases of tissue protrusion, average of the diameter was 1.35mm and average area was 1.03mm$^2$. Proximal edge dissection (n = 40) was more frequent than distal (n = 32), with similar findings of maximal flap opening, maximal flap length, longitudinal flap length and arc of dissection. The intra-observer κ coefficient for SF-OCT was 0.96, and the interobserver κ coefficient was 0.93. The logistic univariate model showed that diabetes mellitus, stent diameter, stent length, proximal reference area, distal reference area, underexpansion and adjuvant procedure were correlated with SF-OCT. The

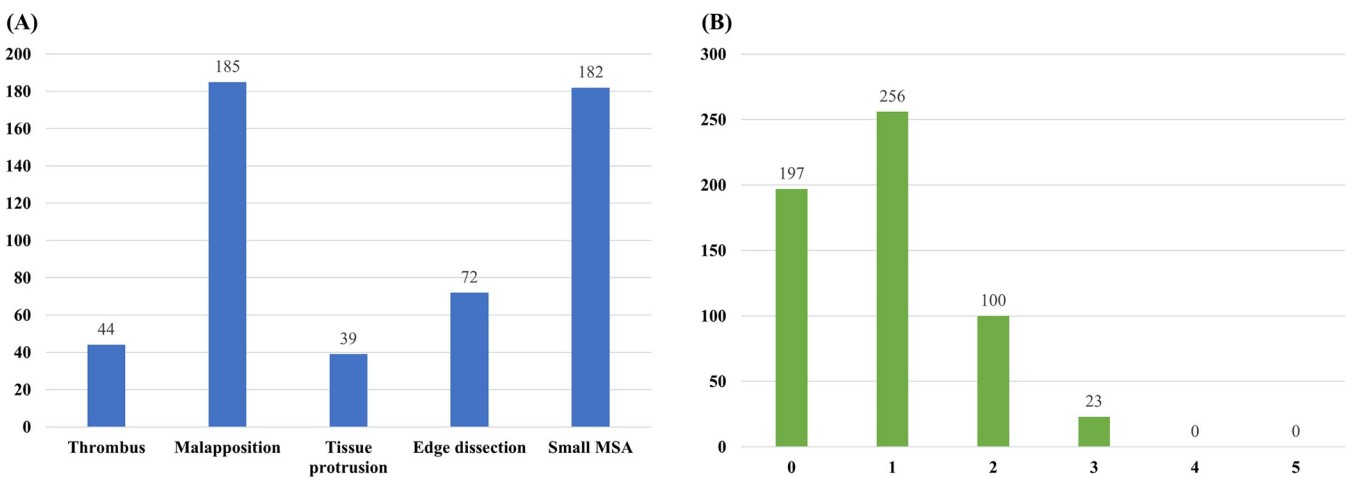

**Fig 3. Incidences of suboptimal findings detected by optical coherence tomography (OCT).** (A) Incidence of each component. (B) Cumulative incidence of suboptimal optical coherence tomography-detected findings.

logistic multivariate model identified stent diameter (OR 0.212; 95% CI 0.136–0.328; P<0.001), and underexpansion (OR 3.244; 95% CI 2.197–4.789; P<0.001) as independent predictors of SF-OCT (Table 4).

## Clinical outcomes

Clinical outcomes on the basis of SF-OCT or No SF-OCT are shown in Table 5. During follow-up (median duration 21.5 months, interquartile range of 15.0–30.0 months), the incidence of MACE was numerically lower in the No SF-OCT group but there was no statistical significance between the two groups (3.0% vs. 5.0%, HR 1.601; 95% CI 0.639–4.011; P = 0.315) (Fig 4). Cox regression analysis was performed to evaluate the predictors of MACE. Overall, neither suboptimal OCT-detected findings nor the components showed significance in univariate analysis (Table 6). In the SF-OCT group, neither each components of SF-OCTs nor the severity of SF-OCTs based on quantitative measurements had statistical power for predicting MACE. Clinical results comparing no adjuvant procedure after OCT (n = 444) versus adjuvant procedures (n = 132) were significantly not different (4.1% vs. 5.3%, HR 1.249; 95% CI 0.521–2.996; P = 0.618).

## Discussion

The main findings of the present study are as follows: (1) SF-OCTs are highly prevalent (65.8%, 379/576 cases) after DES implantation; (2) no SF-OCTs translated into clinical events over the follow-up (median 21.5 months); and (3) independent predictors of SF-OCT were stent diameter and underexpansion. Although several studies have reported that most SF-OCTs after DES implantation resolve spontaneously over a short-term follow-up period, these studies had small sample sizes and did not systematically deal with relevant clinical events [1, 4, 5, 13]. The important differences between the current study and the previous studies are: (1) the present study had the longest follow-up period; (2) the present study compared the clinical outcomes between the SF-OCT group and the No SF-OCT groups; (3) the present study quantitatively analyzed the severity of each component of SF-OCT and the relationship of each component with clinical events; and (4) the present study determined independent factors predictive of suboptimal OCT-detected findings. Additionally, the present study has an

**Table 3. Quantification data of suboptimal optical coherence tomography-detected findings.**

| | Variables | | Average | Q1 | Median | Q3 |
|---|---|---|---|---|---|---|
| Thrombus (n = 116) | Number (n) | | 1.27 | 1 | 1 | 1 |
| | Longitudinal length (mm) | | 0.79 | 0.48 | 0.73 | 1 |
| | Diameter (mm) | | 1.2 | 0.86 | 1.16 | 1.55 |
| | Area (mm$^2$) | | 0.56 | 0.29 | 0.51 | 0.68 |
| Significant thrombus (n = 44) | Number (n) | | 1.05 | 1 | 1 | 1 |
| | Longitudinal length (mm) | | 1.04 | 0.70 | 1.00 | 1.40 |
| | Diameter (mm) | | 1.62 | 1.31 | 1.60 | 1.97 |
| | Area (mm$^2$) | | 0.86 | 0.59 | 0.74 | 1.10 |
| Malapposition (n = 188) | Maximal depth (μm) | | 465 | 310 | 410 | 578 |
| | Area (mm$^2$) | | 2.05 | 1.18 | 1.71 | 2.51 |
| | Length (mm) | | 2.42 | 1.3 | 2.2 | 3.18 |
| Significant malapposition (n = 185) | Maximal depth (μm) | | 469 | 315 | 410 | 580 |
| | Area (mm$^2$) | | 2.07 | 1.21 | 1.72 | 2.53 |
| | Length (mm) | | 2.43 | 1.30 | 2.20 | 3.15 |
| Tissue protrusion (n = 263) | Length (mm) | | 0.89 | 0.62 | 0.84 | 1.1 |
| | Area (mm$^2$) | | 0.37 | 0.16 | 0.25 | 0.42 |
| Significant tissue protrusion (n = 39) | Length (mm) | | 1.35 | 1.05 | 1.30 | 1.67 |
| | Area (mm$^2$) | | 1.03 | 0.56 | 0.69 | 1.02 |
| Edge dissection (n = 100) | Maximal flap opening (mm) | | 0.36 | 0.2 | 0.31 | 0.46 |
| | Maximal flap length (mm) | | 0.8 | 0.38 | 0.7 | 1.02 |
| | Longitudinal flap length (mm) | | 1.73 | 0.93 | 1.5 | 2.3 |
| | Arc (°) | | 28 | 14.8 | 22.3 | 39.5 |
| | Proximal (n = 55) | Maximal flap opening (mm) | 0.4 | 0.2 | 0.33 | 0.52 |
| | | Maximal flap length (mm) | 0.88 | 0.47 | 0.73 | 1.07 |
| | | Longitudinal flap length (mm) | 1.71 | 0.7 | 1.2 | 2.3 |
| | | Arc (°) | 25.2 | 14.2 | 20.3 | 31.9 |
| | Distal (n = 45) | Maximal flap opening (mm) | 0.32 | 0.2 | 0.29 | 0.41 |
| | | Maximal flap length (mm) | 0.69 | 0.32 | 0.58 | 1.01 |
| | | Longitudinal flap length (mm) | 1.74 | 0.25 | 1.8 | 2.35 |
| | | Arc (°) | 33.3 | 18.4 | 31.6 | 48.3 |
| Significant edge dissection (n = 72) | Maximal flap opening (mm) | | 0.44 | 0.27 | 0.37 | 0.52 |
| | Maximal flap length (mm) | | 0.90 | 0.45 | 0.80 | 1.23 |
| | Longitudinal flap length (mm) | | 1.87 | 1.00 | 1.60 | 2.58 |
| | Arc (°) | | 29.1 | 14.2 | 23.9 | 43.0 |
| | Proximal (n = 40) | Maximal flap opening (mm) | 0.49 | 0.28 | 0.38 | 0.71 |
| | | Maximal flap length (mm) | 1.00 | 0.57 | 0.88 | 1.32 |
| | | Longitudinal flap length (mm) | 1.95 | 0.80 | 1.35 | 2.83 |
| | | Arc (°) | 27.6 | 14.2 | 21.9 | 36.9 |
| | Distal (n = 32) | Maximal flap opening (mm) | 0.38 | 0.27 | 0.33 | 0.48 |
| | | Maximal flap length (mm) | 0.78 | 0.40 | 0.69 | 1.18 |
| | | Longitudinal flap length (mm) | 1.78 | 1.30 | 1.80 | 2.30 |
| | | Arc (°) | 32.3 | 15.3 | 34.9 | 47.8 |
| Small MSA (n = 182) | - | - | 3.80 | 3.44 | 3.94 | 4.22 |

Q1: first quartile; Q3: third quartile; MSA: minimal stent area

advantage over previous studies because defining SF-OCT means that these findings do not correlate with clinical events, which can reduce unnecessary additional interventions after

**Table 4. Predictors of suboptimal findings detected by optical coherence tomography.**

| Variable | Univariate analysis | | | | Multivariate analysis | | | |
|---|---|---|---|---|---|---|---|---|
| | OR | 95% CI | | *P*-value | OR | 95% CI | | *P*-value |
| | | Low | High | | | Low | High | |
| Age | 1.006 | 0.991 | 1.022 | 0.414 | | | | |
| Male sex | 0.804 | 0.544 | 1.189 | 0.274 | | | | |
| Unstable angina (vs. stable angina) | 1.269 | 0.843 | 1.908 | 0.253 | | | | |
| MI (vs. stable angina) | 0.975 | 0.629 | 1.511 | 0.910 | | | | |
| Diabetes mellitus | 1.449 | 1.003 | 2.095 | 0.048 | | | | |
| Hypertension | 1.394 | 0.982 | 1.978 | 0.063 | | | | |
| LDL cholesterol | 0.997 | 0.993 | 1.000 | 0.067 | | | | |
| HDL cholesterol | 0.991 | 0.979 | 1.002 | 0.118 | | | | |
| Triglyceride | 1.000 | 0.998 | 1.001 | 0.816 | | | | |
| Peak CK-MB | 1.000 | 0.998 | 1.001 | 0.620 | | | | |
| Stent diameter | 0.212 | 0.139 | 0.324 | <0.001 | 0.212 | 0.136 | 0.328 | <0.001 |
| Stent length | 1.052 | 1.027 | 1.079 | <0.001 | | | | |
| Pre-procedural RD | 0.185 | 0.002 | 17.889 | 0.469 | | | | |
| Pre-procedural MLD | 0.000 | 0.000 | 491.830 | 0.183 | | | | |
| Pre-procedural DS | 0.725 | 0.408 | 1.287 | 0.272 | | | | |
| Pre-procedural lesion length | 3.216 | .999 | 10.353 | 0.050 | | | | |
| Post-procedural RD | 0.978 | 0.014 | 6.670 | 0.104 | | | | |
| Post-procedural MLD | 0.000 | 0.000 | 47.610 | 0.068 | | | | |
| Post-procedural DS | 0.202 | 0.028 | 1.479 | 0.115 | | | | |
| Proximal RA | 0.879 | 0.832 | 0.929 | <0.001 | | | | |
| Distal RA | 0.812 | 0.758 | 0.870 | <0.001 | | | | |
| Underexpansion | 3.238 | 2.240 | 4.681 | <0.001 | 3.244 | 2.197 | 4.789 | <0.001 |
| Adjuvant procedure | 1.447 | 0.944 | 2.216 | <0.001 | | | | |

OR: odds ratio; CI: confidence interval; MI: myocardial infarction; LDL: low-density lipoprotein; HDL: high-density lipoprotein; RD: reference diameter; MLD: minimal lumen diameter; DS: diameter stenosis; RA: reference area

PCI. However, our findings do not signify that intravascular imaging is unnecessary. Stent optimization is a crucial factor for future event prevention and intravascular imaging plays a significant role for correction of suboptimal findings. Even though adjuvant intervention was done after index procedure, underexpansion rate was high, post-procedural MLD was significantly smaller and high rate of SF-OCT was still observed. Since procedure was performed on

**Table 5. Clinical outcomes.**

| Variable | No SF-OCT (197 patients) | SF-OCT (379 patients) | *P*-value |
|---|---|---|---|
| Major cardiovascular adverse event | 6 (3.0) | 19 (5.0) | 0.272 |
| Non-fatal myocardial infarction | 0 | 0 | - |
| All cause death | 1 (0.5) | 2 (0.5) | 0.975 |
| TLR | 4 (2.0) | 13 (3.4) | 0.346 |
| TVR | 5 (2.5) | 17 (4.5) | 0.247 |
| Non-TLR/TVR | 5 (2.5) | 8 (2.1) | 0.743 |
| Stent thrombosis | 1 (0.5) | 1 (0.3) | 0.637 |

SF-OCT: optical coherence tomography detected suboptimal findings; TLR: target lesion revascularization; TVR: target vessel revascularization

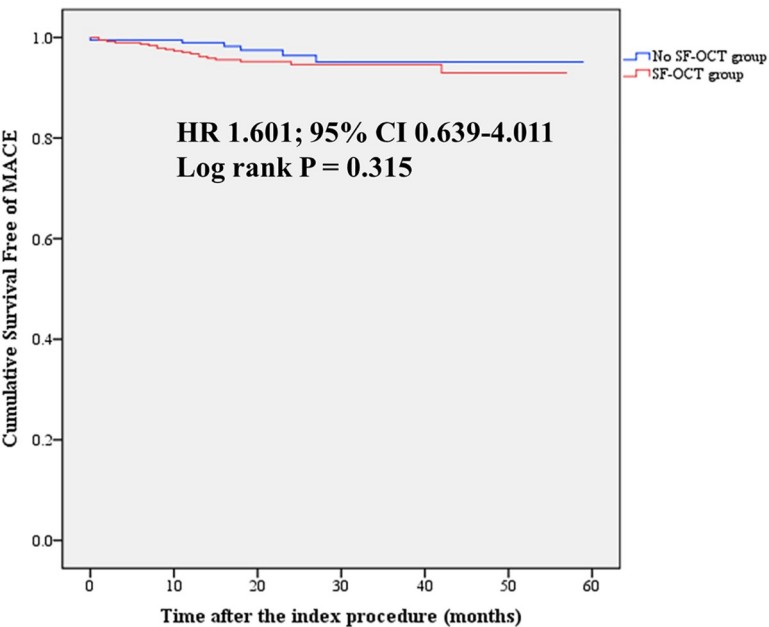

**Fig 4. Survival free of major adverse cardiovascular events according to optical coherence tomography-detected poststent optimal versus suboptimal findings.**

operator's discretion, further management of suboptimal findings was not strict and stent optimization criteria was not firmly determined. Nonetheless, data in current study suggest the concept of safety zone that if the measurements of SF-OCTs are found to similar to our data.

## Malapposition

The incidence of OCT-defined stent malapposition was 32.1% (185/576 lesion) in our study. In studies using intravascular ultrasound (IVUS) for analysis, the incidence of malapposition immediately after stent implantation was approximately 10% [14, 15]. Studies using OCT for analysis have reported higher incidences of post-procedural stent malapposition than studies using IVUS, ranging from 22.2% to more than 50% [1–3, 5]. Since OCT has higher resolution than IVUS, it is superior for detecting malapposition. Our findings also support this high frequency of malapposition. However, this high frequency was not associated with clinical events, a finding similar to those of other previous studies using IVUS and OCT that had follow-up periods less than or approximately one year [1–3, 5, 14, 15]. Previous studies have reported that stent malapposition might be related to late stent thrombosis in both bare-metal stents and DES [16–18]. However, in researches which compared maximal malapposition depth between cases of late stent thrombosis and control showed significantly large scale of malapposition (1400 to 1800 μm) compared to our study (average 469 μm) [17, 18]. Considering the very low incidence of stent thrombosis, the potential impact of malapposition on stent thrombosis can be difficult to evaluate properly. At least, our study can suggest modest degree of malapposition may be left untouched without further correction.

## Thrombus or tissue protrusion

The incidence of OCT-defined significant thrombus was 7.6% (44/576 cases), TP was 6.8% (39/576 cases) and any significant thrombus or TP was 14.2% (82/576 cases) in the present study.

**Table 6. Predictors of major adverse cardiovascular events.**

| Variable | Univariate analysis | | | |
|---|---|---|---|---|
| | HR | 95% CI | | *P*-value |
| | | Low | High | |
| Age | 0.982 | 0.949 | 1.017 | 0.314 |
| Male sex | 1.076 | 0.430 | 2.695 | 0.875 |
| Unstable angina (vs. stable angina) | 0.717 | 0.282 | 1.819 | 0.483 |
| MI (vs. stable angina) | 0.945 | 0.359 | 2.487 | 0.908 |
| Body mass index | 0.972 | 0.824 | 1.146 | 0.734 |
| Diabetes mellitus | 1.161 | 0.521 | 2.588 | 0.715 |
| Hypertension | 0.681 | 0.311 | 1.494 | 0.338 |
| Current smoker | 2.000 | 0.864 | 4.629 | 0.106 |
| Ex-smoker | 0.910 | 0.250 | 3.307 | 0.886 |
| Peak CK-MB | 0.998 | 0.992 | 1.003 | 0.420 |
| hs-CRP | 1.007 | 0.992 | 1.022 | 0.374 |
| LDL cholesterol | 1.000 | 0.993 | 1.008 | 0.942 |
| RAS blocker | 1.014 | 0.459 | 2.239 | 0.973 |
| Beta blocker | 0.589 | 0.263 | 1.315 | 0.196 |
| Stent diameter | 0.937 | 0.382 | 2.301 | 0.887 |
| Stent length | 0.989 | 0.937 | 1.045 | 0.705 |
| MSA | 1.000 | 0.832 | 1.202 | 0.998 |
| Underexpansion | 0.708 | 0.318 | 1.575 | 0.397 |
| Proximal RA | 0.971 | 0.853 | 1.106 | 0.661 |
| Distal RA | 0.989 | 0.852 | 1.148 | 0.885 |
| Expansion rate | 1.014 | 0.978 | 1.052 | 0.438 |
| SF-OCT | 1.601 | 0.639 | 4.011 | 0.315 |
| Thrombus | 6.795 | 0.682 | 67.699 | 0.102 |
| Malapposition | 1.127 | 0.498 | 2.552 | 0.774 |
| Tissue protrusion | 0.037 | 0.000 | 101.382 | 0.414 |
| Edge dissection | 0.568 | 0.134 | 2.411 | 0.443 |
| Small MSA | 1.224 | 0.541 | 2.770 | 0.628 |
| Thrombus diameter | 2.279 | 0.471 | 11.029 | 0.306 |
| Thrombus area | 3.845 | 0.502 | 29.485 | 0.195 |
| Malapposition depth | 0.997 | 0.992 | 1.001 | 0.178 |
| Malapposition area | 0.778 | 0.410 | 1.477 | 0.443 |
| Malapposition length | 1.035 | 0.726 | 1.477 | 0.848 |
| Tissue protrusion length | 1.346 | 0.306 | 5.923 | 0.694 |
| Tissue protrusion area | 0.122 | 0.002 | 8.309 | 0.328 |
| Edge dissection location | 0.671 | 0.061 | 7.412 | 0.745 |
| Edge dissection maximal flap opening | 3.997 | 0.117 | 136.558 | 0.442 |
| Edge dissection maximal flap length | 0.859 | 0.106 | 6.971 | 0.887 |
| Edge dissection longitudinal flap length | 1.550 | 0.908 | 2.646 | 0.108 |
| Edge dissection arc | 0.982 | 0.862 | 1.118 | 0.780 |

HR: hazard ratio; CI: confidence interval; MI: myocardial infarction; hs-CRP: high sensitive C-reactive protein; LDL: low-density lipoprotein; RAS: renin-angiotensin system; MSA: minimal stent area; RA: reference area; SF-OCT: suboptimal findings detected by optical coherence tomography

The rate of thrombus was lower than that in previous reports (37–51%) [1–3]. However, residual thrombus after DES implantation did not translate into clinical events in the current

study. Quantitative thrombus analysis did not reveal any correlations between clinical events and the number of thrombi (Table 6). Only one TVR case showed a thrombus, which was not particularly severe (diameter in the fourth quartile, area in the third quartile). After stent implantation, thrombi are believed to form as a result of procedural problems, such as extended time required for stent location, insufficient heparinization during PCI, or squeezed remnants of pre-existing thrombi after stent implantation [1].

The incidence of TP is quite different from those reported in other OCT studies (81–97.5%) [1–4, 13, 19]. However, not all studies had a high prevalence: e.g., Kume et al. reported a TP incidence of 51.3% [5]. This discordance could be explained by inter-observer bias, or by procedural problems such as high stent balloon pressure or post-balloon pressure. Although quite different frequencies have been observed, most of the OCT studies investigating suboptimal findings concluded that TP appears to be a benign phenomenon. Our data also support these previous results. Soeda et al. reported that irregular protrusion was a powerful predictor of device-oriented clinical endpoints and target lesion revascularization [2]. As regards the definition of TP in this study, our data defined most irregular protrusion as thrombus and only smooth protrusion as TP. However, our study failed to demonstrate that either of the findings was related with clinical outcomes. Through quantification, we found that the average maximal length was 0.87 mm, and the average maximal area was 0.36 mm$^2$. These data are similar to those reported in previous studies [4, 5, 13].

Considering incidence and clinical impact of significant criteria of thrombus or TP, it seems concordance comparing previous studies, which failed to prove significance of these findings [3, 6]. Despite limited sample size and lack of quantification in previous studies, natural course of thrombus and TP were mostly resolved, which can be suggestive explanation of no impact on clinical outcome [1, 4, 5].

## Edge dissection

The incidence of OCT-defined significant ED was 12.5% (72/576 cases) in our study. Compared with other studies, in which the incidence of presence varied widely from 20% to 37.8% and significant criteria of 12% to 14%, our study reported similar frequency [1–3, 6, 8, 20]. Stent ED defined by IVUS or conventional angiography is considered to be associated with increased short-term and mid-term incidences of MACE and stent thrombosis [19, 21–23]. However, we did not find a significant correlation between OCT-defined ED and clinical outcomes after 2 years. Moreover, quantified dissection severity had no clinical impact. The different clinical outcomes might be because of the aim of our study, which was to investigate suboptimal findings in OCT that are apparently normal in angiography. In dissections seen only by OCT that are minor and non-flow-limiting, spontaneous healing might have a benign course and minimal correlation with clinical outcome. Our quantification efforts revealed that the average maximal flap opening was 0.44 mm, the maximal flap length was 0.90 mm, and the average ED longitudinal length was 1.87 mm. In previous studies, the average maximal flap length ranged from 0.7 to 1.0 mm, longitudinal flap length 2.04±1.60mm and the average maximal flap opening was 0.39±0.34 mm [4, 8, 13, 20]. These findings are consistent with those of the present study. These measurement ranges had no clear relationship with any of the clinical outcomes examined.

## Small minimal stent area

Small MSA occupied majority portion among SF-OCTs of 31.6% (182/576 cases). Prevalence was similar compared to previous studies in CLI-OPCI II trial and Soeda et al., which was 23.4% and 41.2% respectively [2, 3]. Previous IVUS as well as OCT studies have shown that

small MSA and inadequate lumen area are associated with major clinical outcomes [2, 3, 24, 25]. Although frequency of small MSA did not differ in this study, it was not associated with clinical events like other studies. Reasons of this discrepancy can be considered by dominant use of new generation stent and high rate of dual antiplatelet therapy, as well as limited number of patients enrolled in current study.

## Study limitations

The major limitation of this study is the absence of follow-up OCT data to assess the natural healing course of SF-OCTs. However, the most important issue regarding these findings is whether they have a significant clinical impact that requires additional initial procedures. Another concern is this was a non-randomized retrospective study based on low event rates, relatively limited sample size and modest follow-up period to clarify the clinical outcome, raising the possibility of selection bias and therefore underpowered to determine the benefits of correcting SF-OCTs. Estimating predicted power of suboptimal findings in clinical events were performed with power analysis of 0.553, which this study may be underpowered for its primary endpoint. Since the data collection of clinical outcomes was based on retrospective chart review, there is a chance of underreporting. The dataset used in the study is not recent and therefore results can be underestimated. However, data in current study suggest the concept of safety zone since certain findings in OCT to or not to intervene is still yet controversial and our data can at least show that modest degree can be tolerated. Only the maximum depth and maximum malapposition area were measured, and the entire stent was not evaluated. Finally, all measurements were performed manually, meaning that a certain degree of manual error might be present. Larger studies with a longer follow-up duration are needed to confirm the relationships between clinical events and SF-OCTs.

## Conclusions

The presence of angiographically insignificant SF-OCTs (ED, TP, malapposition, thrombus and small MSA) and their severity were not associated with clinical outcomes in this study.

## Supporting information

**S1 Dataset.**
(XLSX)

## Author Contributions

**Conceptualization:** Cheol Woong Yu.

**Data curation:** Jae Young Cho, Hyungdon Kook, Cheol Woong Yu.

**Formal analysis:** Jae Young Cho, Hyungdon Kook, Cheol Woong Yu.

**Funding acquisition:** Jae Young Cho.

**Validation:** Jae Young Cho, Hyungdon Kook.

**Writing – original draft:** Jae Young Cho, Hyungdon Kook.

**Writing – review & editing:** Cheol Woong Yu.

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
