## [Decision Letter · Decision Letter 0]

10 Aug 2020

PONE-D-20-16120

Clinical impact of angiographically insignificant suboptimal poststent findings detected by optical coherence tomography after drug-eluting stent implantation

PLOS ONE

Dear Dr. Yu,

Thank you for submitting your manuscript to PLOS ONE. After careful consideration, we feel that it has merit but does not fully meet PLOS ONE’s publication criteria as it currently stands. Therefore, we invite you to submit a revised version of the manuscript that addresses the points raised during the review process.

Generally, the reviewers found the paper favorable, however there are some issues that need to be addressed. Please address each item specifically and outline these changes in revised text and in a point by point fashion in the response letter.

We look forward to receiving your revised manuscript.

Kind regards,

Jay Widmer

Academic Editor

PLOS ONE

Journal Requirements:

"This study was supported by Wonkwang University in 2018."

"The authors received no specific funding for this work."

Reviewers' comments:

Reviewer's Responses to Questions

**Comments to the Author**

1. Is the manuscript technically sound, and do the data support the conclusions?

Reviewer #1: Yes

Reviewer #2: Partly

2. Has the statistical analysis been performed appropriately and rigorously? 

Reviewer #1: I Don't Know

Reviewer #2: Yes

3. Have the authors made all data underlying the findings in their manuscript fully available?

Reviewer #1: Yes

Reviewer #2: No

4. Is the manuscript presented in an intelligible fashion and written in standard English?

Reviewer #1: Yes

Reviewer #2: Yes

5. Review Comments to the Author

Reviewer #1: Interesting and important question to investigate. Study is reasonably designed. Size is modest, although it could be underpowered. Evidence of this includes a HR for the primary endpoint of 1.6 (0.6-4), suggesting a sizable effect could be present (in a larger sample). Also suggestion of under-powering is evident in the fact that statistically significant differences in MSA and other such measures (reference vessel size, etc) did not translate into difference in restenosis/TVR. Are we to conclude that final stent size does not impact restenosis?

Questions that arise include:

Why were all patients treated between Jan 2011 - May 2013, and the analysis just now being performed? Was the database cut off, and we have no more info, or was there some reason that this time period was chosen? The data is getting to be fairly old, and of course, techniques evolve which could impact how we apply findings.

I'd like to understand a bit better what the operators knew about and did with the OCT results. They seemed to have them available, and yet sometimes they were still considered "suboptimal." Was this because the operator mis-interpreted the OCT images, had no option because there was no solution to the issue, or was it simply an issue that what the operator deemed insignificant or suboptimal was different from the later, agreed upon definition. Nonetheless, it points out a significant weak point in the study, this is not a randomized comparison of one strategy vs. another, rather it is a comparison of a stent procedure with optimal results vs. a procedure with suboptimal results, but one in which the operator deemed the results worthy to accept. That is an important distinction.

I like and appreciate the suggestion of a "safety zone" in optimizing PCI results. The study contradicts what the majority of studies (and meta-analyses) have shown, that some of these suboptimal results do negatively impact outcomes. That being said, it is likely that as the author purport, some degree of suboptimal results can be tolerated without compromising clinical results. How much can be tolerated, is a fascinating question.

Reviewer #2: This study sought to compare clinical outcomes may differ between patients with SF (suboptimal findings) OCT and without SF-OCTs after DES implantation. With this aim 576 with final OCT after DES were divided into SF-OCT group (n=379, 379 lesions) and No SF-OCT group (n=197, 197 lesions). The study population had no significant abnormal finding in final angiography. Among 379 patients with SF-OCT, 32.4% had multiple SF-OCTs. Malapposition (32.1%) was the most frequent, followed by small MSA (31.6%), edge dissection (12.5%), thrombus (7.6%) and tissue protrusion (6.8%). These features were also quantitatively analyzed. The SF-OCT group showed smaller ST diameter and longer length, and lower expansion rate. The incidence of MACE did not differ between the two groups (3% vs 5%, p=0.310).

This is an interesting study suggesting that suboptimal OCT findings have no clinical consequences in patients with good angiographic results.

Some issues, however, need to be addressed:

1. Sample size remains a major problem in this study (unpowered) as clinical events in patients with good angiographic results after DES are low. Please provide an estimate of the predicted power of the study and acknowledge that the study was unpowered for its primary endpoint.

2. Study criteria should be clarified. 2) Patients who underwent sequential OCT immediately after DES implantation or after adjuvant procedures; 3) No significant abnormal finding in final coronary angiography. Adjuvant procedures after OCT examination were performed in 132 patients (132 lesions). Were clinical results poorer in patients that required optimization of the stent after the initial OCT?

3. Information of clinical outcomes was collected by the retrospective review of the chart. Was this performed blindly to angiographic data and OCT findings?

4. Angiographic MLD after the procedure was smaller in the SF-OCT group. This is a classical variable associated with poorer long-term clinical and angiographic outcomes. Again a lack of power may explain the lack of clinical differences between the groups.

5. The logistic multivariate model identified stent diameter MSA and underexpansion

as independent predictors of SF-OCT. The variable MSA by OCT is a factor already included in the SF-OCT list.

6. Only clear abnormal findings were factors were considered “significant” SF-OCTs. Some abnormal findings were considered SF-OCT. However cualitative crietria were arbitrary. Perhaps a different cut-offs would yield different results. Please address. Among them it is included a small MSA defined as in-stent minimum area <4.5 mm2

MSA has been found to predict clinical outcomes (mainly TLR and stent thrombosis) in many previous IVUS and OCT studies. Please address. Again the study may be simply unpowered in this regard.

7. Please describe with further details the QCA analysis and the system used. The reference diameter, minimal luminal diameter, percentage of stenosis, and lesion length were evaluated. Were measurements performed also after DES implantation? Any of the findings described in the SF OCT group related to poorer angiographic findings by QCA?

6. PLOS authors have the option to publish the peer review history of their article (what does this mean?). If published, this will include your full peer review and any attached files.

Reviewer #1: **Yes: **Timothy A. Mixon MD, Daniel Fronk MD.

Reviewer #2: No

---

## [Author Response · Author response to Decision Letter 0]

18 Sep 2020

Ref: Manuscript Number PONE-D-20-16120

Clinical impact of angiographically insignificant suboptimal poststent findings detected by optical coherence tomography after drug-eluting stent implantation

Answers to the editor’s and reviewers’ comments 

We thank the editor and the reviewers for their thoughtful and constructive comments on our study. Several issues were raised by the editor and reviewers, and we have addressed these issues point-by-point and enclosed our comments. Indeed, we realized that the comments were insightful, constructive, and helpful in strengthening our manuscript. We have used red color font to indicate the revised portions of our manuscript for the reviewers. We hope that the editor and the reviewers would be satisfied with our responses and find the revised manuscript suitable for publication in PLOS ONE.

Response to Reviewer #1:

Comment #1.

Interesting and important question to investigate. Study is reasonably designed. Size is modest, although it could be underpowered. Evidence of this includes a HR for the primary endpoint of 1.6 (0.6-4), suggesting a sizable effect could be present (in a larger sample). Also suggestion of under-powering is evident in the fact that statistically significant differences in MSA and other such measures (reference vessel size, etc) did not translate into difference in restenosis/TVR. Are we to conclude that final stent size does not impact restenosis?

Answer:

Thank you for your valuable comment we should address. As the reviewer pointed out, the small sample size is one of the important limitations of this study. However, what differentiates this study from previous studies is that the present study was conducted on patients who showed only suboptimal findings confined to OCT, where no definite abnormal finding was observed on coronary angiography. Therefore, the subjects of this study have a lower frequency of event occurrence in clinical outcomes than other studies. As a result, it is estimated that small MSA or underexpansion, which have been known as risk factors of the adverse clinical outcomes, did not translate into difference in clinical outcomes in this study. In addition, the fact that this study included patients who had only single vessel to intervene may be an additional explanation for the results that did not show significant difference in clinical outcomes between the two study groups. As the reviewer points out, in general, the final stent size is considered a factor that affects restenosis. However, it would be reasonable to interpret that a slight difference in stent size does not directly lead to a difference in clinical outcomes in patients with simple lesions and no definite angiographic abnormal findings. We believe that the conclusions of this study should be reconfirmed in future large-scale studies. The limitations of this study on small sample size are specified in the Limitation section.

In page 25 line 9-14:

Another concern is this was a non-randomized retrospective study based on low event rates, relatively limited sample size and modest follow-up period to clarify the clinical outcome, raising the possibility of selection bias and therefore underpowered to determine the benefits of correcting SF-OCTs. Estimating predicted power of suboptimal findings in clinical events were performed with power analysis of 0.553, which this study may be underpowered for its primary endpoint.

In page 25 line 21-22:

Larger studies with a longer follow-up duration are needed to confirm the relationships between clinical events and SF-OCTs.

Comment #2.

Why were all patients treated between Jan 2011 - May 2013, and the analysis just now being performed? Was the database cut off, and we have no more info, or was there some reason that this time period was chosen? The data is getting to be fairly old, and of course, techniques evolve which could impact how we apply findings.

Answer:

Thank you for your valid comments. We agree that the period of dataset is not recent. The first announcement of present study date was made in TCT 2014, and then TCT 2015 and TCT 2016. 

JACC Volume 64, Issue 11 Supplement, September 2014 (DOI: 10.1016/j.jacc.2014.07.426)

JACC Volume 66, Issue 15 Supplement, October 2015 (DOI: 10.1016/j.jacc.2015.08.976)

JACC Volume 68, Issue 18 Supplement, November 2016 (DOI: 10.1016/j.jacc.2016.09.697)

Since then, considerable time has been spent due to repeated intravascular image analysis and statistical analysis to increase the precision of the data. The point you have mentioned has been added in the Limitation section.

In page 25 line 15 to 16:

The dataset used in the study is not recent and therefore results can be underestimated. 

Comment #3:

I'd like to understand a bit better what the operators knew about and did with the OCT results. They seemed to have them available, and yet sometimes they were still considered "suboptimal." Was this because the operator mis-interpreted the OCT images, had no option because there was no solution to the issue, or was it simply an issue that what the operator deemed insignificant or suboptimal was different from the later, agreed upon definition. Nonetheless, it points out a significant weak point in the study, this is not a randomized comparison of one strategy vs. another, rather it is a comparison of a stent procedure with optimal results vs. a procedure with suboptimal results, but one in which the operator deemed the results worthy to accept. That is an important distinction.

Answer:

Thank you for providing these insights. Since procedure was performed on operator’s personal discretion, further management of suboptimal findings was not strict and stent optimization criteria was not firmly determined. Brief OCT analysis was performed in on-site and concurrent analysis in core-lab was not done. In addition, because the resolution of OCT is very high, there are often cases in real world practice where there is a sub-optimal finding on OCT that did not show any specific findings on coronary angiography. In these cases, there are many cases of contemplating whether it is beneficial to apply an additional procedure to correct suboptimal findings on OCT without specific findings on coronary angiography. This is because, by applying an additional procedure, we may get a chance to correct the suboptimal finding on the OCT, but on the contrary, there is also a risk of complications such as dissection or need for additional stent implantation by performing additional procedures to correct suboptimal findings of OCT. We believe this study is valuable in that it can provide some modest additional information to these common situations that can be encountered in many medical centers using OCT. The point you have mentioned has been added in the manuscript.

In page 21 line 18 to 22:

Even though adjuvant intervention was done after index procedure, underexpansion rate was high, post-procedural MLD was significantly smaller and high rate of SF-OCT was still observed. Since procedure was performed on operator’s discretion, further management of suboptimal findings was not strict and stent optimization criteria was not firmly determined.

Comment #4:

I like and appreciate the suggestion of a "safety zone" in optimizing PCI results. The study contradicts what the majority of studies (and meta-analyses) have shown, that some of these suboptimal results do negatively impact outcomes. That being said, it is likely that as the author purport, some degree of suboptimal results can be tolerated without compromising clinical results. How much can be tolerated, is a fascinating question.

Answer:

Thank you for your suggestion and we agree with your assessment. Even though under expansion rate was very high, our data did not show association with clinical outcome, unlike previous studies about suboptimal findings after PCI. In guidelines and results of large randomized studies tell us about importance of intravascular imaging for stent optimization, however, not all procedures undergo intravascular imaging in real world for several reasons (e.g., financial problem, lack of time, operator’s personal discretion, and impossible optimization due to lesion characteristics). Even though intravascular imaging was done, on-site precise measuring and analysis for optimization is quite burden. Certain findings in OCT to or not to intervene is still yet controversial and our data can at least show that modest degree can be tolerated. However, although this study provided the concept of "safety zone" in optimizing PCI results, it is difficult to accurately define what the size of the safety zone is only with the results of this study. Based on the results of this study, a large-scale OCT study should be conducted to clarify the boundaries of the safety zone. Thank you again for your valuable comments.

In page 25 line 16 to 18:

However, data in current study suggest the concept of safety zone since certain findings in OCT to or not to intervene is still yet controversial and our data can at least show that modest degree can be tolerated.

 

Response to Reviewer #2:

Comment #1:

Sample size remains a major problem in this study (unpowered) as clinical events in patients with good angiographic results after DES are low. Please provide an estimate of the predicted power of the study and acknowledge that the study was unpowered for its primary endpoint.

Answer:

Thank you for your sharp comments. Since present study was based on retrospective registry, estimating sample size was not possibly calculated. Although the follow-up period of this study was relatively long compared to other OCT studies, the small number of patients with clinical events due to the nature of the study population demographics (had only single vessel to intervene, no definite abnormal finding in final angiography) is an important limitation of this study. Estimating predicted power of suboptimal findings in clinical events were performed with power analysis of 0.553, which this study may be underpowered for its primary endpoint as you mentioned. The point you have mentioned has been added in the Limitation section.

In page 25 line 12 to 14:

Estimating predicted power of suboptimal findings in clinical events were performed with power analysis of 0.553, which this study may be underpowered for its primary endpoint.

Comment #2:

Study criteria should be clarified. 2) Patients who underwent sequential OCT immediately after DES implantation or after adjuvant procedures; 3) No significant abnormal finding in final coronary angiography. Adjuvant procedures after OCT examination were performed in 132 patients (132 lesions). Were clinical results poorer in patients that required optimization of the stent after the initial OCT?

Answer:

Thank you for providing the insight. Clinical results comparing no adjuvant procedure (n=444) versus adjuvant procedures (n=132) were significantly not different (4.1% vs. 5.3%, HR 1.249; 95% CI 0.521-2.996; P = 0.618). However, the lack of clinical difference between these two groups may be due to the fact that only patients with no definite abnormal findings in the final coronary angiography after additional procedures were included in this study. Therefore, it should not be interpreted that the implementation of the additional procedure does not affect clinical outcomes. This finding is addressed in the manuscript.

In page 18 line 6 to 8:

Clinical results comparing no adjuvant procedure (n=444) versus adjuvant procedures (n=132) were significantly not different (4.1% vs. 5.3%, HR 1.249; 95% CI 0.521-2.996; P = 0.618).

Comment #3:

Information of clinical outcomes was collected by the retrospective review of the chart. Was this performed blindly to angiographic data and OCT findings?

Answer:

Thanks for your kind advice. Collecting data of clinical information and outcomes was performed blindly to angiographic data and OCT findings. The point you mentioned has been added in the manuscript.

In page 5 line 23 to 24:

Collecting data of clinical information and outcomes was performed blindly to angiographic data and OCT findings.

Comment #4:

Angiographic MLD after the procedure was smaller in the SF-OCT group. This is a classical variable associated with poorer long-term clinical and angiographic outcomes. Again a lack of power may explain the lack of clinical differences between the groups.

Answer:

Thank you for your comment and we agree with your assessment. Even though underexpansion rate was high and angiographic MLD after the procedure was significantly smaller in the SF-OCT group, our data did not show association with clinical outcome. As you mentioned above, small sample size and lack of power can be the reason of this finding. However, what differentiates this study from previous studies is that the present study was conducted on patients who showed only suboptimal findings confined to OCT, where no definite abnormal finding was observed on coronary angiography. Therefore, the subjects of this study had a lower frequency of event occurrence in clinical outcomes than other studies. As a result, it is estimated that small MLD or MSA, which were known as risk factors of the adverse clinical outcomes, did not translate into difference in clinical outcomes in this study. In addition, the fact that this study included patients who had only single vessel to intervene may be an additional explanation for the results that did not show significant difference in clinical outcomes between the two study groups. It would be reasonable to interpret that slight differences in MLD (MLD 3.35 ± 0.44 vs. 3.00 ± 0.46mm, p=0.015), which are guaranteed to be extended to some extent (mean value of MLD in both group≥3.0mm), may not directly lead to a difference in clinical outcomes in patients with simple lesions and no definite angiographic abnormal findings. This has been pointed out in the manuscript.

In page 21 line 18-20:

Even though adjuvant intervention was done after index procedure, underexpansion rate was high, post-procedural MLD was significantly smaller and high rate of SF-OCT was still observed.

Comment #5:

The logistic multivariate model identified stent diameter MSA and underexpansion as independent predictors of SF-OCT. The variable MSA by OCT is a factor already included in the SF-OCT list.

Answer:

Thanks for your sharp point. The variable MSA is deleted and analyzed again in logistic multivariate model of independent predictors for SF-OCT (Table 4).

In page 13 line 20 to page 14 line 2:

The logistic univariate model showed that diabetes mellitus, stent diameter, stent length, proximal reference area, distal reference area, underexpansion and adjuvant procedure were correlated with SF-OCT. The logistic multivariate model identified stent diameter (OR 0.212; 95% CI 0.136-0.328; P<0.001), and underexpansion (OR 3.244; 95% CI 2.197-4.789; P<0.001) as independent predictors of SF-OCT (Table 4).

In page 16 to 17 Table 4:

Variable Univariate analysis Multivariate analysis

 OR 95% CI P-value OR 95% CI P-value

 Low High Low High 

Age 1.006 0.991 1.022 0.414 

Male sex 0.804 0.544 1.189 0.274 

Unstable angina (vs. stable angina) 1.269 0.843 1.908 0.253 

MI (vs. stable angina) 0.975 0.629 1.511 0.910 

Diabetes mellitus 1.449 1.003 2.095 0.048 

Hypertension 1.394 0.982 1.978 0.063 

LDL cholesterol 0.997 0.993 1.000 0.067 

HDL cholesterol 0.991 0.979 1.002 0.118 

Triglyceride 1.000 0.998 1.001 0.816 

Peak CK-MB 1.000 0.998 1.001 0.620 

Stent diameter 0.212 0.139 0.324 <0.001 0.212 0.136 0.328 <0.001

Stent length 1.052 1.027 1.079 <0.001 

Pre-procedural RD 0.185 0.002 17.889 0.469 

Pre-procedural MLD 0.000 0.000 491.830 0.183 

Pre-procedural DS 0.725 0.408 1.287 0.272 

Pre-procedural lesion length 3.216 .999 10.353 0.050 

Post-procedural RD 0.978 0.014 6.670 0.104 

Post-procedural MLD 0.000 0.000 47.610 0.068 

Post-procedural DS 0.202 0.028 1.479 0.115 

Proximal RA 0.879 0.832 0.929 <0.001 

Distal RA 0.812 0.758 0.870 <0.001 

Underexpansion 3.238 2.240 4.681 <0.001 3.244 2.197 4.789 <0.001

Adjuvant procedure 1.447 0.944 2.216 <0.001 

Comment #6:

Only clear abnormal findings were factors were considered “significant” SF-OCTs. Some abnormal findings were considered SF-OCT. However qualitative criteria were arbitrary. Perhaps a different cut-offs would yield different results. Please address. 

Among them it is included a small MSA defined as in-stent minimum area <4.5 mm2. MSA has been found to predict clinical outcomes (mainly TLR and stent thrombosis) in many previous IVUS and OCT studies. Please address. Again the study may be simply unpowered in this regard.

Answer:

You have asked an important question that we should address. Measurement of SF-OCTs and “significant” SF-OCTs are separately addressed in the quantification data of SF-OCT (Table 3). All statistical analysis was performed on basis of “significant” SF-OCTs.

Average (3.80), median (3.94), Q1 (3.44) and Q3 (4.22) value of MSA in small MSA group is added in the quantification data of SF-OCT (Table 3).

For the reason why small MSA did not affect clinical outcome in this study, it would be appreciated if you refer to the answer to Comment #4.

In page 14 to 16 Table 3:

 Variables Average Q1 Median Q3

Thrombus

(n=116) Number (n) 1.27 1 1 1

 Longitudinal length (mm) 0.79 0.48 0.73 1

 Diameter (mm) 1.2 0.86 1.16 1.55

 Area (mm2) 0.56 0.29 0.51 0.68

Significant Thrombus

(n=44) Number (n) 1.05 1 1 1

 Longitudinal length (mm) 1.04 0.70 1.00 1.40

 Diameter (mm) 1.62 1.31 1.60 1.97

 Area (mm2) 0.86 0.59 0.74 1.10

Malapposition

(n=188) Maximal depth (µm) 465 310 410 578

 Area (mm2) 2.05 1.18 1.71 2.51

 Length (mm) 2.42 1.3 2.2 3.18

Significant Malapposition

(n=185) Maximal depth (µm) 469 315 410 580

 Area (mm2) 2.07 1.21 1.72 2.53

 Length (mm) 2.43 1.30 2.20 3.15

Tissue protrusion

(n=263) Length (mm) 0.89 0.62 0.84 1.1

 Area (mm2) 0.37 0.16 0.25 0.42

Significant Tissue protrusion

(n=39) Length (mm) 1.35 1.05 1.30 1.67

 Area (mm2) 1.03 0.56 0.69 1.02

Edge dissection

(n=100) Maximal flap opening (mm) 0.36 0.2 0.31 0.46

 Maximal flap length (mm) 0.8 0.38 0.7 1.02

 Longitudinal flap length (mm) 1.73 0.93 1.5 2.3

 Arc (°) 28 14.8 22.3 39.5

 Proximal

(n=55) Maximal flap opening (mm) 0.4 0.2 0.33 0.52

 Maximal flap length (mm) 0.88 0.47 0.73 1.07

 Longitudinal flap length (mm) 1.71 0.7 1.2 2.3

 Arc (°) 25.2 14.2 20.3 31.9

 Distal

(n=45) Maximal flap opening (mm) 0.32 0.2 0.29 0.41

 Maximal flap length (mm) 0.69 0.32 0.58 1.01

 Longitudinal flap length (mm) 1.74 0.25 1.8 2.35

 Arc (°) 33.3 18.4 31.6 48.3

Significant Edge dissection

(n=72) Maximal flap opening (mm) 0.44 0.27 0.37 0.52

 Maximal flap length (mm) 0.90 0.45 0.80 1.23

 Longitudinal flap length (mm) 1.87 1.00 1.60 2.58

 Arc (°) 29.1 14.2 23.9 43.0

 Proximal (n=40) Maximal flap opening (mm) 0.49 0.28 0.38 0.71

 Maximal flap length (mm) 1.00 0.57 0.88 1.32

 Longitudinal flap length (mm) 1.95 0.80 1.35 2.83

 Arc (°) 27.6 14.2 21.9 36.9

 Distal (n=32) Maximal flap opening (mm) 0.38 0.27 0.33 0.48

 Maximal flap length (mm) 0.78 0.40 0.69 1.18

 Longitudinal flap length (mm) 1.78 1.30 1.80 2.30

 Arc (°) 32.3 15.3 34.9 47.8

Small MSA

(n=182) - - 3.80 3.44 3.94 4.22

Comment #7:

Please describe with further details the QCA analysis and the system used. The reference diameter, minimal luminal diameter, percentage of stenosis, and lesion length were evaluated. Were measurements performed also after DES implantation? Any of the findings described in the SF OCT group related to poorer angiographic findings by QCA?

Answer:

Thank you very much for your valuable comments. A detailed explanation of QCA analysis is provided in the Methods section. Measurements of QCA analysis after DES implantation is presented in angiographic and procedural data (Table 2). Measurements of QCA analysis were analyzed in logistic multivariate model of independent predictors for SF-OCT, and none of those measurements were related to SF-OCTs. These findings are address in the predictors of SF-OCTs (Table 4).

In page 6 line 5-10:

Coronary angiograms were analyzed using a computer-based Telecardiology system, version 2.02 (Medcon Inc., Tel Aviv, Israel) by three radiologic technicians who were blinded to the study purpose. The reference diameter, minimal luminal diameter (MLD), percentage of stenosis, and lesion length were evaluated from diastolic frames using guided catheter magnification calibration in a single, matched view with a computerized quantitative analyzer using a caliper.

In page 16 to 17 Table 4:

Variable Univariate analysis Multivariate analysis

 OR 95% CI P-value OR 95% CI P-value

 Low High Low High 

Pre-procedural RD 0.185 0.002 17.889 0.469 

Pre-procedural MLD 0.000 0.000 491.830 0.183 

Pre-procedural DS 0.725 0.408 1.287 0.272 

Pre-procedural lesion length 3.216 .999 10.353 0.050 

Post-procedural RD 0.978 0.014 6.670 0.104 

Post-procedural MLD 0.000 0.000 47.610 0.068 

Post-procedural DS 0.202 0.028 1.479 0.115

---

## [Decision Letter · Decision Letter 1]

5 Oct 2020

Clinical impact of angiographically insignificant suboptimal poststent findings detected by optical coherence tomography after drug-eluting stent implantation

PONE-D-20-16120R1

Dear Dr. Yu,

We’re pleased to inform you that your manuscript has been judged scientifically suitable for publication and will be formally accepted for publication once it meets all outstanding technical requirements.

Kind regards,

Jay Widmer

Academic Editor

PLOS ONE

Reviewer's Responses to Questions

**Comments to the Author**

1. If the authors have adequately addressed your comments raised in a previous round of review and you feel that this manuscript is now acceptable for publication, you may indicate that here to bypass the “Comments to the Author” section, enter your conflict of interest statement in the “Confidential to Editor” section, and submit your "Accept" recommendation.

Reviewer #1: All comments have been addressed

Reviewer #2: All comments have been addressed

2. Is the manuscript technically sound, and do the data support the conclusions?

Reviewer #1: Yes

Reviewer #2: Partly

3. Has the statistical analysis been performed appropriately and rigorously? 

Reviewer #1: Yes

Reviewer #2: Yes

4. Have the authors made all data underlying the findings in their manuscript fully available?

Reviewer #1: Yes

Reviewer #2: Yes

5. Is the manuscript presented in an intelligible fashion and written in standard English?

Reviewer #1: No

Reviewer #2: Yes

6. Review Comments to the Author

Reviewer #1: Thank you for addressing my concerns and questions. Some minor alterations are needed which are mainly language issues, including:

Page 5, line 24: "was performed in a blinded fashion in regards to angiographic data and OCt..."

P 18, line 7: "change to "not significantly different"

P 21, line 18: "Even though adjuvant intervention was performed after the index procedure, the rate of under expansion was high...and a high rate of SF-OCT was still observed"

p21, line 20-24: "Since the procedure was performed at the operator's discretion, further management of suboptimal findings was not based on strict protocol, and stent optimization criteria were not universally followed.

Reviewer #2: (No Response)

7. PLOS authors have the option to publish the peer review history of their article (what does this mean?). If published, this will include your full peer review and any attached files.

Reviewer #1: No

Reviewer #2: **Yes: **Fernando Alfonso MD

---

## [Editor Report · Acceptance letter]

8 Oct 2020

PONE-D-20-16120R1 

Clinical impact of angiographically insignificant suboptimal poststent findings detected by optical coherence tomography after drug-eluting stent implantation 

Dear Dr. Yu:

I'm pleased to inform you that your manuscript has been deemed suitable for publication in PLOS ONE. Congratulations! Your manuscript is now with our production department. 

Kind regards, 

on behalf of

Dr. Jay Widmer 

Academic Editor

PLOS ONE